# Bacterial Nanocellulose Hydrogel: A Promising Alternative Material for the Fabrication of Engineered Vascular Grafts

**DOI:** 10.3390/polym15183812

**Published:** 2023-09-18

**Authors:** Daichen Liu, Qingshan Meng, Jinguang Hu

**Affiliations:** Department of Chemical and Petroleum Engineering, University of Calgary, 2500 University Drive, Calgary, AB T2N 1N4, Canada; daichen.liu@ucalgary.ca (D.L.); qingshan.meng@ucalgary.ca (Q.M.)

**Keywords:** bacterial nanocellulose, vascular grafts, natural hydrogel

## Abstract

Blood vessels are crucial in the human body, providing essential nutrients to all tissues while facilitating waste removal. As the incidence of cardiovascular disease rises, the demand for efficient treatments increases concurrently. Currently, the predominant interventions for cardiovascular disease are autografts and allografts. Although effective, they present limitations including high costs and inconsistent success rates. Recently, synthetic vascular grafts, made from artificial materials, have emerged as promising alternatives to traditional methods. Among these materials, bacterial cellulose hydrogel exhibits significant potential for tissue engineering applications, particularly in developing nanoscale platforms that regulate cell behavior and promote tissue regeneration, attributed to its notable physicochemical and biocompatible properties. This study reviews recent progress in fabricating engineered vascular grafts using bacterial nanocellulose, demonstrating the efficacy of bacterial cellulose hydrogel as a biomaterial for synthetic vascular grafts, specifically for stimulating angiogenesis and neovascularization.

## 1. Introduction

Blood vessels are vital conduits for nutrient supply and waste elimination in all tissues [1]. They pervade nearly all tissues and organs, especially those metabolically active, such as the liver and heart. The advancement of artificial blood vessels is spurred primarily by two factors.

Firstly, the growing number of cardiovascular disease patients, with over 600,000 vascular implants reported annually in the United States alone, highlights the need. The global vascular grafts market, valued at USD 5.37 billion in 2021, is projected to reach USD 8.5 billion by 2030 (Figure 1). With the increasing incidence of cardiovascular diseases, this market is expected to grow at a Compound Annual Growth Rate (CAGR) of 5.24% from 2022 to 2030.

The typical treatment for cardiovascular diseases involves replacing narrowed or blocked vessels. Two prevalent vascular grafting methods exist: vascular autografts, derived directly from the patient’s vessels, and artificial blood vessels, usually fabricated from synthetic materials such as expanded polytetrafluoroethylene (ePTFE, i.e., Gortex) and poly(ethylene terephthalate) (PET, i.e., Dacron) [2,3]. Autografts boast excellent adaptability and patency but may cause harm to some patients when harvested. Artificial vessels have been used for large-diameter blood vessel replacements since 1956, as they exhibit negligible thrombosis due to high blood flow and low resistance [4,5]. However, they demonstrate low patency in small diameter vessels (<6 mm), with patency rates reported at 40% at 6 months and 25% at 3 years [6]. Therefore, polymers developed for small-diameter vessel synthesis face stringent criteria, including processability, mechanical properties, morphology, porosity, surface wettability, and biocompatibility.

Secondly, tissue engineering, a burgeoning field, grapples with a similar issue: the lack of reliable artificial blood vessels. Tissue engineers aspire to create in vitro cultures of large organs, far beyond two-dimensional cell cultures and miniature three-dimensional tissue organs. However, the success of tissue engineering hinges on the supply of essential nutrients, oxygen, and bioactive signals to cells through blood vessels [7,8,9]. For tissues larger than a few hundred microns, the diffusion limit of most biomolecules is exceeded, necessitating a robust vascular network to maintain tissue viability and functionality. When artificial tissues are implanted in the body, the inflammation response to wound healing releases hypoxia-induced angiogenic growth factor (GF), triggering spontaneous angiogenesis in the tissue [9]. However, this activity can only provide a vascular growth rate of a few tenths of a micron per day, which may lead to hypoxic cell death in the tissue interior [10]. Hence, a functional vascularity or vascular network is essential for successful tissue engineering.

To address these pressing issues, researchers have turned their attention towards bacterial cellulose. As a highly pure form of crystalline cellulose produced by bacteria, it is considered superior to plant and wood-derived cellulose, making it one of the most abundant biocompatible materials on Earth [11]. In addition, the abundance of hydroxyl groups in BC also facilitates chemical modification, and its 3D structure permits physical modification with other polymers and nanomaterials. This adaptability allows researchers to enhance or add desired properties to BC, broadening its potential for biomedical applications. Thus, bacterial cellulose (BC) has seen extensive application in biomedicine, including wound healing, drug delivery, tissue engineering, and biosensors, due to its remarkable properties and easy modification. 

Bacterial cellulose is composed of ribbon-like protofibers, each less than 100 nm in width, which are assembled from basic nanofibers measuring 7–8 nm in width (Figure 2A). This distinctive architecture grants bacterial cellulose notable properties such as crystallinity, flexibility, and exceptional purity, free from lignin or hemicellulose. Thus, it serves as an ideal candidate for developing robust and flexible tissue engineering materials across diverse scales [12,13,14]. After undergoing lyophilization, bacterial cellulose retains its unique three-dimensional structure (Figure 2B). Moreover, it can be manipulated to form a miniature tubular structure (Figure 2C), effectively mimicking a blood vessel’s structure. This property establishes its aptitude for use in vascular scaffold applications. Furthermore, bacterial cellulose’s highly porous structure coupled with its high-water absorption capacity, similar to the human body’s natural extracellular matrix, facilitates cell adhesion and proliferation. Consequently, it is a favored material for the construction of vascular scaffolds [15,16]. This study will summarize the developments and applications of bacterial cellulose (BC) in creating artificial blood vessels, including its physical properties, biochemistry, and modifications, along with prospective highlights.

## 2. Review of Conventional Materials Used in Vascular Grafts

The inherent physical and biochemical properties of bacterial cellulose position it as a promising candidate for synthetic blood vessel scaffolding materials [18]. However, before elaborating on its unique attributes, it is essential to explore the evolution of artificial blood vessel development.

Human bodies, despite their susceptibility to damage, possess an astounding capacity for self-repair. As biomaterials and regenerative medicine evolve, the idea of replacing damaged tissue is inching closer to reality. Modern vascular techniques owe their genesis to the groundbreaking work of Alexis Carrel, a French surgeon, who established a method for anatomizing natural blood vessels in the early 20th century, an achievement that won him the Nobel Prize in Physiology or Medicine in 1912 [19]. In 1966, Charles Sparks introduced silicone rubber as a material for arterial replacements [20]. Yet, crafting fully artificial blood vessels populated with cells remains challenging. 

Cells in culture, while proliferative, struggle to form intricate three-dimensional structures; this is where scaffolds come into play. Vascular engineering employs synthetic or natural biofriendly materials to fabricate scaffolds, providing a framework for cellular attachment and growth. Biofriendly materials—derived from living organisms or those that demonstrate biocompatibility—are designed to emulate the host tissue they are intended to replace [21,22,23]. 

The journey towards crafting tissue-engineered blood vessels has witnessed significant milestones since Weinberg and Bell’s pioneering effort in 1986. They developed a tubular tissue structure using bovine vascular smooth muscle cells, endothelial cells, and fibroblasts. However, the first model suffered from a few limitations, such as a low tensile strength of approximately 90 mm Hg, necessitating additional support from a Dacron cuff [24]. The following years saw the contributions of many researchers, pushing the boundaries of vascular engineering. For instance, Zilla’s team developed a method for seeding the lumen of a polytetrafluoroethylene (ePTFE) tube with endothelial cells [25], while Langer’s team used polyglycolic acid (PGA) scaffolds for cell culture [26,27]. L’Heureux’s team developed a unique method using fibroblasts and vascular smooth muscle cells from human exosomal sources, and in the 21st century, Hoerstrup’s team advanced the field with a scaffold made of PGA–P4HB, which closely resembled a human pulmonary artery [28]. To more concretely visualize this progression, refer to Figure 3. It provides a glimpse into the early stages of artificial blood vessel development. Although not reflecting specific milestone events, the images and descriptions showcase the various strategies explored over time. From early experiments with silicone rubber to the more recent uses of polytetrafluoroethylene (ePTFE) and polyglycolic acid (PGA), this journey illustrates the evolution and improvement of materials and techniques used in vascular engineering. It demonstrates how far the field has come and lays the foundation for understanding why bacterial cellulose has emerged as a promising material for vascular grafts. Each iteration in the development of artificial blood vessels has presented its unique challenges and breakthroughs, ultimately leading to the current focus on bacterial cellulose and its potential to surpass the limitations identified with these earlier materials. 

As our understanding of human-compatible materials has advanced, porous materials have emerged as a promising avenue to emulate the extracellular matrix (ECM) and promote dynamic cell growth. Scaffolds facilitate vital support for cells, allowing them to thrive and function, thus creating an optimal environment for cell proliferation [29,30,31]. To mimic protein–cell interactions at a molecular level, scaffolds are designed to provide chemical or structural cues, such as incorporating growth factors and peptide sequences such as arginyl–glycyl–aspartate (RGD) [32,33]. Figure 4 is a visual representation of the evolution in materials used for vascular grafts, progressing from autologous arteries or veins to artificial blood vessels, and finally to the current focus of research—biosynthetic small-diameter vessels, such as those fabricated from bacterial cellulose. This diagram provides a clear comparison of the advantages and disadvantages associated with each type of material, highlighting the need for improved solutions in the field of vascular grafts.

Technology has enabled an increase in the use of large and medium-diameter artificial vessels in various procedures, boasting a patency rate of about 90% when replacing aortoiliac, carotid, and common femoral arteries [34]. However, smaller-diameter vessels present a unique challenge, with autologous veins being the primary source for these vessels. Autologous saphenous veins, due to their accessibility and favorable conditions, serve as an ideal source for small and medium-diameter vessels in surgical procedures. Nonetheless, their procurement amplifies surgical trauma, and their limited availability further exacerbates the issue [35,36,37,38,39]. With an ever-growing clinical demand, the current sources of autologous vessels fall short, creating an urgent need to explore reliable alternative sources. This search is paramount to minimize surgical impact and to ensure successful patient outcomes.

In light of the limitations of autologous grafts, synthetic materials have gained considerable attention as an alternative. Among synthetic grafts, expanded polytetrafluoroethylene (ePTFE) and polyethylene terephthalate (PET, Dacron) grafts are commonly used for larger diameter vessels. While these synthetic materials have shown reasonable long-term patency rates for large diameter grafts (>6 mm), they perform poorly when used for small-diameter (<6 mm) vascular grafts due to thrombosis and intimal hyperplasia [2,3]. The quest for better synthetic alternatives has prompted exploration into the potential of bioresorbable polymers, such as polyglycolic acid (PGA), polylactic acid (PLA), and poly(lactic-co-glycolic) acid (PLGA), which can degrade over time to minimize foreign body response. However, these materials also face challenges, including loss of mechanical strength during degradation and acidic degradation products, which may lead to inflammation [40,41,42,43].

In addition to synthetic materials, biological materials such as collagen, fibrin and decellularized tissue matrices have been investigated for vascular graft applications. These materials often possess superior biocompatibility, but their low mechanical strength and potential for immunogenicity present hurdles to their widespread clinical use [44,45,46,47,48].

In summary, while significant progress has been made in the development of vascular grafts, there is no ‘one-size-fits-all’ solution. The choice of material for a vascular graft depends on a multitude of factors including the size of the graft, the location of implantation, the patient’s health condition, and more. There is a continuous need for the development of advanced materials that can closely mimic the properties of natural blood vessels, resist thrombosis, promote cell integration, and withstand the mechanical stresses of the circulatory system. This is where bacterial nanocellulose hydrogel exhibits its promise, offering the potential to overcome many of these challenges and serve as an ideal material for the fabrication of engineered vascular grafts. 

## 3. Fabrication and Modification Techniques of Bacterial Nanocellulose Hydrogels

Despite extensive efforts, conventional materials and technologies have yet to fully address the challenges associated with the development of small-diameter artificial blood vessels. Bacterial cellulose, in particular, has emerged as a promising candidate due to its unique properties, such as high purity, excellent mechanical strength, impressive water-holding capacity, and distinctive nanostructured network. These features set it apart from other alternatives and have attracted significant attention from the scientific community. 

Natural blood vessels, with their intricate layers and physiological functions, set a high standard for any synthetic substitute [49,50]. Therefore, the primary focus of current research is to create structures that closely resemble these natural blood vessels, leveraging our ever-evolving understanding and technological advancements. Through thoughtful design and surface modification, bacterial cellulose scaffolds can provide ideal porosity, viscoelastic responses, and surface topography to enhance cell attachment. Furthermore, various fabrication methods for bacterial cellulose tubes have been developed, enabling the achievement of mechanical properties that meet or even surpass those of natural blood vessels.

However, despite the significant strides made in scaffold design, several scientific challenges still need to be addressed to create the ideal scaffold. Here, the biological properties of bacterial cellulose, including its biocompatibility with the human body, its mechanical and morphological characteristics under varying culture conditions, and its non-toxicity towards endothelial cells will be critically reviewed in this following section [14,51,52]. Strategies for addressing challenges are also highlighted.

### 3.1. Production of Bacterial Cellulose

Bacterial cellulose (BC), first identified in the edible Philippine product “nata-de-coco”, boasts a rich history dating back to its formal classification as cellulose in 1886 by Adrian Brown. This was precipitated by his observations of a film formation during the fermentation of acetic acid with a bacterium, initially known as *Acetobacter aceti*. The bacterium, which has had multiple names throughout history, such as *Acetobacter xylinum* and *Gluconacetobacter xylinus*, is currently recognized as *Komagataeibacter xylinus* [53]. 

Researchers have strived to develop efficient BC culture methods, building on the primitive techniques initially employed in the Philippines. Their endeavors aim at enhancing yield while simultaneously reducing costs. Key discoveries in this effort reveal that different carbon and nitrogen sources would greatly influence BC’s properties, including water retention capacity, crystallinity, molecular weight, and polymerization. *K. xylinus* has demonstrated the capability to utilize a broad spectrum of carbon sources, ranging from glucose, fructose, glycerol, and sucrose, to industrial and agricultural waste such as rice straw and corn pulp. As indicated in Table 1 below, the BC yield varies noticeably depending on the carbon source used.

The Standard medium was used as a control and contained 2.0% glucose, 0.5% yeast extract, 0.5% polypeptone, 0.675% Na_2_HPO_4_·12H_2_O, and 0.115% citric acid monohydrate in distilled water (pH 6.0). The production medium was modified from the standard medium and used to investigate the effect of different carbon sources on cellulose production. Various carbon sources were provided at 2% (*w*/*v* or *v*/*v*) instead of glucose as the carbon source in the standard medium. The optimized medium was established based on the results of the study. The optimum medium composition was 4% glucose, 0.1% yeast extract, 0.7% polypeptone, and 0.8% Na_2_HPO_4_·12H_2_O. The addition of ethanol to the optimized medium enhanced cellulose production. In an optimized medium containing 1.4% (*v*/*v*) ethanol, cellulose production was 15.2 g/L, which was about four times higher than that without ethanol.

Raw materials significantly contribute to bacterial growth by providing not only carbon sources but also other essential factors, including nitrogen sources. For instance, the use of rice bark as a raw material supplies vital vitamins and nitrogen necessary for BC production. Altering the feedstock in BC production not only affects bacterial growth but also modifies BC’s physical properties. Experiments demonstrate that the crystal plane of BC can transition from type I to type II cellulose when rice bark is used as a raw material [61]. Interestingly, ethanol serves dual roles during BC production and fermentation, acting as a carbon source and reducing the mutation rate from BC-producing strains to non-BC-producing strains [53]. Despite these findings, the Hestrin–Schramm medium (H–S medium), comprising a mixture of glucose and yeast extract, is the preferred choice for artificial vascular research. This is attributed to its cost-effectiveness, simplicity, and consistent production of high-quality BC. The glucose and yeast extract concentrations can be modulated to tailor the properties of the resulting BC, enhancing its suitability for specific applications. 

Understanding the influence of different raw materials on the production of BC provides a crucial foundation for optimizing its yields. Equally important, however, is a deep understanding of the biochemistry underpinning BC synthesis. This is because the synthesis process is directly impacted by the nutrient composition of the growth medium, which in turn influences the characteristics of the produced BC. In this next section, synthetic mechanisms of BC in *K. xylinus* will be interpreted, offering insights into the metabolic pathways that transform these raw materials into BC.

### 3.2. The Synthetic Mechanisms of BC in K. xylinus

The synthesis of bacterial cellulose (BC) by *K. xylinus* can be delineated into four principal stages: (1)conversion of glucose into glucose-6-phosphate.(2)transformation of glucose-6-phosphate into glucose-1-phosphate, catalyzed by glucose phosphate mutase.(3)conversion of glucose-1-phosphate into UDP-glucose under the action of uridine diphosphate glucose (UDP-glucose) pyrophosphorylase.(4)final conversion of UDP-glucose into cellulose.

The initiation of BC formation involves the metabolic pool of hexose phosphate. When *K. xylinus* employs glucose as its carbon source for BC synthesis, a minor proportion of the glucose is directly phosphorylated to yield glucose-6-phosphate, with the majority being oxidized to produce extracellular gluconate. The hexose phosphate pool is subsequently replenished through the pentose cycle and the gluconeogenesis pathway.

Research by Liu et al. suggests that enhancing the metabolic pool of hexose phosphate can substantially boost BC production. This augmentation was accomplished by disrupting the glucose dehydrogenase gene (gcd) of *K. xylinus*, and overexpressing both the glucose facilitator protein gene (glf) and the endogenous glucose kinase gene (glk). With this expanded hexose phosphate pool, the BC yield of *K. xylinus* attained a remarkable 52% under static fermentation conditions [15,53]. Figure 5 provides a schematic representation of cellulose production by bacteria.

### 3.3. Mechanical Strength of BC Tubes

After thoroughly investigating the synthetic pathways of BC in *K. xylinus*, it is essential to consider the mechanical strength of the resulting BC tubes, given the pressures of blood flow and blood pressure they must endure. This strength is a multifactorial attribute, deriving from parts such as biomaterial selection, scaffold design, and cellular metabolism [62,63]. Of note, a standout characteristic of BC is the in-situ plasticity, which enables bacteria to form cellulose films at the interface between the growth medium and air during production. This unique property enables the seamless shaping of BC using permeable molds, which consequently enhances mechanical strength.

Dieter Klemm leveraged this characteristic to produce BC tubes of various diameters (BASYC^®^-tube) using *A. xylium*. He demonstrated through subsequent tests that these small diameter BC tubes can withstand a radial force of 800 mN and a pressure of at least 20 kPa [64]. This approach is reflected in Figure 6D, which shows BASYC^®^-tubes with different inner diameters, wall thicknesses, and lengths, and Figure 6E, a schematic representation of the tension test used to measure their mechanical strength. Instead of using this characteristic, Henrik Bäckdahl and colleagues employed a circular ring with a 10 mm outer diameter and 5 mm inner diameter, cut from the BC after it had developed into a substantial membrane [65]. This method of culturing BC tubes under different diameter molds is represented in Figure 6F. Ananda Putra and colleagues employed a silicone tube as a mold and found that its shape influenced the alignment of BC fibers. They noted that when the silicone tube’s inner diameter was below 8 mm, the BC fiber pairs tended to align axially, thus boosting the axial loading capacity of the BC tube while reducing its radial loading capacity [66]. 

Hong et al. innovatively used bilayer silica tubes and blended Kombucha with *A. xylinum*, an approach represented in Figure 6B. This synergistic approach accelerated BC production, resulting in thicker BC tubes with a substantially reduced production time. Using this combination, they were able to produce BC tubes with a 3 mm wall thickness in just 7 days, compared to 25 days when using *A. xylinum* alone [66]. Meanwhile, Dr. Corzo Salinas utilized polydimethylsiloxane (PDMS) as a mold to investigate the effect of incubation time on BC performance, as shown in Figure 6C. They discovered that as incubation time increased, so too did the mechanical properties of the BC tubes [67].

**Figure 6 polymers-15-03812-f006:**
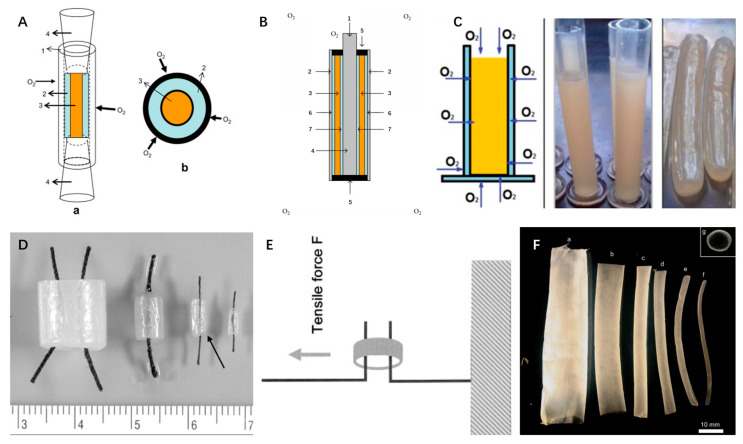
This figure presents various schematic diagrams and results related to the production and testing of bacterial cellulose (BC) tubes. (**A**) Illustration of the use of a single layer of silicone tubing, where the tube is filled with medium and BC is grown between the medium and oxygen, (1) wall of the silicone tube; (2) bacterial cellulose; (3) culture medium; (4) plug [66]; (**B**) Depiction of a coaxial layout of double-layered silicone tubes, where the mid-layer is filled with medium to grow thicker BC tubes, Oxygen enters from 1&2; 3 is the BC medium; 4 is the central ventilation duct; 5 represents the upper and lower lids of the device; 6 and 7 are the BCs growing on the inner and outer sides, respectively [68]; (**C**) The use of a single-layer PDSM mold to culture BC tubes, operating on a principle similar to that in A [67]; (**D**) BASYC^®^-tubes of varying inside diameters, wall thicknesses, and lengths, demonstrating their suitability for experimental microsurgical applications [64]; (**E**) A schematic representation of a tension test, illustrating the mechanical testing of the BC tubes [64]; (**F**) BC tubes grown under molds of different diameters [57]. This figure highlights the versatility of BC and the various methods employed for its cultivation and testing in the context of vascular graft engineering.

In another experiment, the resistance of bacterial cellulose tubes was affected by varying the concentration of oxygen during incubation with bacterial cellulose, i.e., concentrations such as 21% (air), 35%, 50% and 100%. The mechanical properties of the pipes, such as burst pressure and tensile properties, were also evaluated. The burst pressure of the BC tube increased with increasing oxygen/gas ratio, reaching a maximum of 880 mmHg at 100% oxygen [51]. Table 2 presents a summary of the variations in bacterial cellulose strength under different culture conditions. The parameters evaluated include the temperature at which the cultures were maintained, the duration of the culture period, the medium used for the cultures, and the type of mold employed.

In specific terms, this table includes details such as the inner diameter (ID) and outer diameter (OD) or wall thickness of the resultant bacterial cellulose. Furthermore, maximum radial and axial loads or stress exerted on the bacterial cellulose under these conditions are provided. 

This data emphasizes the versatility of bacterial cellulose, pointing to its potential for customization according to specific requirements. This adaptability makes bacterial cellulose an exciting material to explore the applications that require tunable mechanical properties include burst pressure, tensile properties, compressive strength etc.

In conclusion, the mechanical strength of bacterial cellulose is a dynamic attribute, considerably influenced by a myriad of factors including bacterial strain, culture conditions, and physical parameters of the molds used. Researchers have demonstrated the ability to manipulate these factors, resulting in BC structures with a broad spectrum of mechanical properties. This adaptability not only underscores the versatility of BC as a biomaterial but also paves the way for its customization based on specific application needs.

As we progress further into the exploration of BC as a biomaterial for vascular grafts, it becomes crucial to evaluate another vital aspect of its applicability: biocompatibility. This aspect, central to the success of any biomaterial used within the body, will be examined thoroughly in the next section, ensuring that we leave no stone unturned in our comprehensive review of BC’s potential as a material for vascular grafts.

### 3.4. Biocompatibility of BC

Biocompatibility is a crucial factor in assessing the viability of artificial stents. Ideally, a graft’s design and composition should elicit minimal immune response, thereby mitigating immune cell recruitment. Biocompatible grafts can mitigate immune-mediated reactions and inflammation, which might otherwise lead to cell death and exacerbate pathological outcomes. Thus, a scaffold must integrate with host tissue without instigating detrimental immune responses [69,70]. Due to its outstanding biocompatibility, BC has found extensive use in therapeutic, regenerative medicine, and other related fields. Applications of BC include periodontal tissue regeneration barriers, BC dressings for chronic wounds and burn wounds, bone tissue scaffolds, and nerve scaffolds, among others [71].

In this paragraph, we will specifically discuss the compatibility of BC with vascular tissues and blood. Pioneering the use of BC in vascular implants, Dieter Klemm introduced Bacterial synthesized cellulose (BASYC) as an artificial blood vessel material in 2001. Using *Bacillus cereus*, he created vascular implants with high mechanical strength, excellent water retention, low internal surface roughness, and superior biocompatibility [56]. These BASYC^®^ tubes, with an internal diameter of about 1 mm, were successfully implanted in the carotid artery of white rats, maintaining a 100% patency rate during the 4-week study period with no signs of clotting or proliferation of the inner wall of blood vessels [64]. Building on Klemm’s work, Dieter A. Schumann and colleagues conducted further biocompatibility and vascular patency tests on BC implants in rat and porcine carotid arteries. They reported excellent integration with no thrombosis observed after a year in rats and an 87.5% 3-month co-natal rate in porcine subjects [72]. Scherner and Malm’s separate studies on sheep further corroborated BC’s biocompatibility. They both achieved a 50% patency rate over a 3-month period after replacing sheep carotid arteries with BC tubes. Notably, Scherner observed a good luminal endothelialization process, and Malm reported patent BC tubes in two sheep, even 13 months post-surgery [73,74]. A comparative study by Fink et al. highlighted BC’s lower thrombosis rate compared to commonly used vascular materials such as ePTFE and PET, thereby validating its potential as an excellent material for artificial blood vessels [75].

To further evaluate its compatibility with human cells, scientists conducted long-term in vivo tests and blood compatibility assays, preceding attempts to transplant vascular cells onto BC. These cell culture tests used venous endothelial cells, smooth muscle cells, and fibroblasts, which originate from the inner, middle, and outer layers of blood vessels. In the development of BC vascular grafts, it is crucial to understand the native architecture of blood vessels. Figure 7 provides a detailed schematic of both large and small blood vessels’ anatomy, illustrating the structural diversity inherent in our vascular system. Specifically, it showcases the three main layers: tunica intima, tunica media, and tunica adventitia.

#### 3.4.1. Tunica Intima (Inner Layer)

The tunica intima, the innermost vascular wall layer, consists primarily of endothelial cells (ECs). These specialized cells, as depicted in Figure 7, maintain laminar blood flow, prevent intravascular thrombosis, and participate in adaptive immune responses [76,77,78]. An interesting experiment was conducted by Shanshan Zang and colleagues wherein they inoculated BC tubes with all three types of vascular cells and assessed their growth using the CCK-8 kit. The findings highlighted that while fibroblasts completed their life cycle—from initiation to apoptosis—both smooth muscle cells (SMCs) and human umbilical vein endothelial cells (HUVECs) reached a growth plateau. This pattern further affirmed the non-toxic nature of BC tubes for vascular cells. Focusing particularly on HUVECs, the researchers observed vigorous growth, characterized by normal morphology and a clear distinction between the nucleus and cytoplasm. The cells took on a plate-like spindle or polygonal shape, forming a mosaic or cobblestone monolayer arrangement. These observations hint at the excellent biocompatibility of BC tubes with HUVECs, thereby emphasizing BC’s potential as a viable material for tissue engineering and regenerative medicine applications [79].

#### 3.4.2. Tunica Media (Middle Layer)

The tunica media, or the middle layer of the vascular wall, primarily comprises smooth muscle cells (SMCs) arranged in a circular pattern also shown in Figure 7. Their essential functions include maintaining the vascular wall’s structure and regulating vascular tone to ensure adequate intravascular pressure and tissue perfusion. In situations of vessel damage, SMCs release platelet-derived growth factor (PDGF) to promote their own proliferation and assist in thrombus formation by endothelial cells (ECs), which aids wound healing [76,80]. Additionally, SMCs secrete collagen fibers, elastic fibers, elastic laminae, and proteoglycans that contribute to maintaining vascular elasticity and radial compliance [78]. Notably, an experimental investigation led by Henrik Bäckdahl demonstrated that SMCs could adhere to and grow on static BC. The SMCs showcased a promising growth of 40 µm inwards within a fortnight [65].

#### 3.4.3. Tunica Adventitia (Outer Layer)

The tunica adventitia, or the outer layer of the vascular wall, is mainly made up of connective tissue—including collagen fiber bundles, an elastic fiber network, fibroblasts—as well as the vasa vasorum and nervi vasorum. This layer performs numerous crucial functions, such as connecting arteries to surrounding tissues, maintaining vascular smooth muscle cells’ proper tone, and regulating local blood pressure through the innervation of the vessel wall as shown in Figure 7. Additionally, it plays a vital role in immune cell recruitment and the production of inflammatory factors to counter vascular diseases. The tunica adventitia also serves as a protective layer and lends support to the vascular wall’s underlying layers [81]. However, a study conducted by Neeracha Sanchavanakit et al. indicated that unmodified BC might not optimally promote fibroblast growth, spread, and migration. While BC was not toxic to fibroblasts, it did not as effectively facilitate their growth as other materials [82]. Thus, while BC has shown significant potential for encouraging certain vascular cells’ growth and proliferation, its suitability for promoting fibroblast activity needs further exploration.

The outlined experiments underscore BC’s potential as a vascular scaffold due to its ability to support certain types of vascular cells’ growth and proliferation. Nonetheless, BC also presents challenges that require further research for complete resolution. For instance, BC’s prolonged patency rate in animal experiments ranged from 50% to 87.5%, suggesting a need for further optimization. Additionally, while BC’s clotting rate is slower than materials such as ePTFE and PET, it may still cause clotting, leading to blockages or thrombi. Lastly, despite BC’s non-cytotoxic nature, its lack of cell adhesion hinders cell growth and tissue regeneration. 

**Figure 7 polymers-15-03812-f007:**
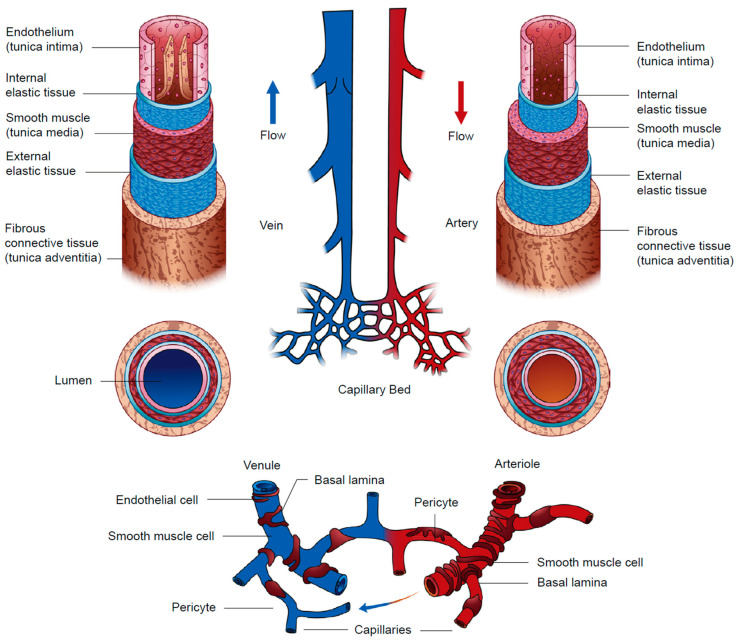
This figure provides a comprehensive representation of the structural composition of both arterial and venous vessels. It illustrates the different layers and components, including the tunica intima, tunica media, and tunica adventitia. The diagram also details the specific characteristics of large and small vessels, providing an understanding of the diversity in vascular architecture. This figure is adapted from Stratman et al. [83].

By aiming to mimic this native structure with BC-based vascular grafts, the objective is to fabricate a construct that can seamlessly integrate with the body’s existing vasculature and fulfill the desired functions with minimal complications. To accomplish this, scientists have recognized the need to optimize bacterial cellulose to ameliorate its shortcomings, such as inadequate cell adhesion. In the subsequent sections, we will delve into the physical and chemical strategies that researchers have adopted to enhance the properties of bacterial cellulose. The discussion will specifically concentrate on factors such as pore size, porosity, surface characteristics, and functionalized scaffolds.

### 3.5. Pore Size and Porosity

Artificial scaffolds significantly contribute to uniform cell distribution and growth, necessitating the presence of interconnected porous cavities. These facilitate the establishment and expansion of micro-vessels within the scaffold [69,84]. The interconnectivity and lumen density of the scaffold enable the exchange of nutrients, gases, and efficient waste removal by cells [85]. The role of pore size is also critical in determining the load capacity of the scaffold, as seen in Figure 8. Panel A presents a schematic diagram of the fermentation unit where the difference in buoyancy between paraffin and starch granules is utilized to generate a tangled network of bacterial cellulose [86]. A smaller pore size may impede the cell infiltration pathway, reducing the total cell density per unit volume of the graft and hampering ECM production and vascular infiltration into the construct [87,88]. As demonstrated in the SEM images (Figure 8B), the use of porosity agents such as potato starch and paraffin significantly affect the pore structures of the BC tubes [86]. However, this may also produce excessively large pore sizes, hindering cell migration into the material [13]. Furthermore, the creation of porous bacterial cellulose is feasible, with porosity agents such as salt, paraffin, ice, gelatin, and sugar being employed in various polymeric materials, including BC. A comparative study used potato starch and paraffin as porosity agents for BC tubes. The paraffin particles of varying sizes contributed to cellulose pore creation. Subjecting these particles to a specific temperature allowed the partially melted paraffin to modify cellulose pore size and interconnectivity. Successful smooth muscle cell adherence within these pores was also observed [85,87,88].

The drying process, which significantly influences BC properties such as morphology, porosity, and mechanical strength, is also important to consider. As shown in Figure 8C, the freeze-drying method can greatly influence the porous and fibrous nature of BC, depending on the solution used [89]. BC is a hydrogel with high water content, and its fibrous mesh holds over 99% water [90], the drying method can substantially affect BC properties such as morphology, porosity, and mechanical strength. Common methods include ambient or air drying and oven drying, both of which may cause the wet material’s original porous microstructure to collapse, reducing its porosity [91]. In contrast, freeze-drying, which involves freezing the water and then sublimating it [92], preserves the fiber’s morphology and porosity [86]. Studies by Zeng et al. demonstrated that the crystallinity of BC films dried at room temperature was lower than that of freeze-dried (FD) films. Moreover, fiber diameter exhibited minimal change during the freeze-drying processes [91]. Further research by Xiong et al. used freeze-drying to fabricate porous 3D BC scaffolds and examined their influence on human breast cancer cell behavior. The scaffolds provided an enhanced environment for cell adhesion, growth, proliferation, and infiltration, as verified by proliferation, histological and scanning electron microscopy (SEM) analyses [89]. Likewise, Hu et al. fabricated 3D BC using a combination of acetic acid treatment and freeze-drying, achieving high porosity and exceptional human fibroblast cell adhesion and infiltration [93]. Freeze-drying maintains the fibrous and porous characteristics of native BC, offering sites for drug loading, liquid and gas absorption, and nanomaterial attachment and impregnation.

With a clear understanding of how pore size and porosity impact the performance of BC-based vascular grafts, the focus now shifts to another crucial aspect—the surface properties of the scaffolds. The surface properties, which include surface topography and chemical characteristics, have a profound effect on cellular interactions and thus significantly influence the performance of the scaffold. The upcoming section provides a deep dive into the various strategies used to optimize these properties.

### 3.6. Modification and Functionalization of BC

The surface properties of a vascular graft, such as topography and chemical characteristics, have a profound impact on cellular attachment and proliferation [94]. It is noteworthy that cellular adhesion to synthetic surfaces hinges on the surface’s chemical attributes. Furthermore, surface morphology can induce cells to adopt an arrangement reminiscent of specific tissues, thereby regulating differentiation rates [95,96,97]. The properties of the inner surface of a graft are of utmost importance since the material directly contacts the blood and therefore, must exhibit antithrombogenic properties to prevent clot formation. In natural vasculature, endothelial cells display a broad spectrum of functions and adaptations. Consequently, researchers are exploring methods to adhere an intact endothelial layer onto the interior surface of vascular substitutes. This creates a smooth, thrombus-resistant surface, which, while not a perfect replica, provides functions similar to those of the ECM within the human body. To inhibit clot formation, modifying the scaffold surface with antithrombotic substances, such as heparin, is an essential consideration in vascular tissue engineering. Engineered vascular grafts must exhibit antithrombogenic properties to deter platelet activation and subsequent thrombus formation, which can lead to vessel blockage. Further surface modifications, such as the application of peptides such as RGD (Arg–Gly–Asp), can enhance endothelial cell attachment to the scaffold surface, acting as an antithrombotic barrier. A promising strategy is to chemically modify or integrate the adhesive properties of cells with the structure of bacterial cellulose to facilitate adherence to a cellulose scaffold [65,98,99,100,101,102]. 

Helen Fink and Fábia K demonstrated increased cell adhesion by employing a xylan (XG) glycoconjugation approach, which attached the cell adhesion peptide RGD to bacterial cellulose [101]. Others have used chimeric proteins containing cellulose binding modules (CBM) in conjunction with RGD to bind RGD molecules to bacterial cellulose [98]. Beyond RGD, a variety of modification techniques have been explored by researchers. For example, treating bacterial cellulose/fibrin composites with glutaraldehyde can enhance their mechanical properties to more closely resemble those of natural small-diameter blood vessels [100]. Immobilizing heparin on the surface of bacterial cellulose or bacterial cellulose composites through cross-linking techniques has demonstrated improved hemocompatibility of BC by promoting endothelial cell growth [102,103,104].

Magnetic bacterial cellulose (MBC) has been developed by incorporating Fe_3_O_4_ (magnetite) nanoparticles, which enhance the rapid retention of smooth muscle cells in damaged vascular systems and promote reperfusion [105]. Lei Zhang and his team advanced this approach by integrating MBC with Arg–Gly–Asp (RGD) active peptide compounds (RMBC). The resultant structure was subjected to an oscillating magnetic field to regulate cell adhesion and endothelialization [106]. Moreover, Li Liu and colleagues synthesized a hierarchical structure of bacterial cellulose/potato starch (BC/PS) composites by adding solubilized PS. The resulting material featured a dense inner surface and an outer layer with large circular pores, which promoted in vivo cell adhesion. This led to the formation of intact endothelial monolayers, organized smooth muscle cells, abundant neocapillaries, and extracellular matrix deposition [107]. This composite represents a promising progress in surface modification of BC grafts.

Having discussed the various optimization methods and surface modifications of BC in the previous sections, it becomes evident that significant progress has been made in tailoring these scaffolds to mimic native vascular structures. However, the complexity of human vasculature presents ongoing challenges. The next frontier in this field involves not just mimicking the structure of blood vessels, but also integrating multiple cell types and complex architectures. Additionally, advanced strategies employed in engineering complex vascular structures, as well as the use of innovative technologies such as 3D printing and the incorporation of smart materials will be discussed in the following section. It explores how these advancements take us a step closer to developing fully functional, biocompatible, and biodegradable vascular grafts.

## 4. Construction of Complex Vascular Structures

One of the significant challenges in vascular tissue engineering is the selection of an optimal cell source. Autologous material typically tops the preference list, given its potential to minimize the risk of immunological reactions. However, the procurement of adequate cellular resources is not always feasible. As a solution, the exploitation of stem cells from autologous, allogenic, or xenogeneic sources has been suggested, given their ability to differentiate into diverse cell types. In an effort to emulate the intricate three-layered vascular cell structures or dendritic vascular structures in the human body, researchers have innovatively employed bacterial cellulose. For instance, Ying Li et al. [108] designed a bacterial cellulose film with shape memory and self-rolling properties. The film was seeded and cultured with three types of vascular cells—human umbilical vein endothelial cells (HUVEC), human aortic smooth muscle cells (HASMC), and human scleral fibroblasts (HSF). Upon completion of the culture process, the cellulose film self-rolled, culminating in a complex, in vitro three-layered vascular structure. 

To generate intricate tree-like structures, Sanna Sämfors et al. [109] utilized polylactic acid (PLA) to create a sacrificial scaffold with a convoluted lumen structure using 3D printing technology. This scaffold was then enveloped by growing bacterial cellulose in a bioculture. Post growth, the PLA scaffold was hydrolyzed and removed during bacterial cellulose purification using NaOH, thereby producing a complex dendritic vascular structure composed of bacterial cellulose. Following the inoculation and culture of endothelial cells within the tube, a sophisticated endothelial dendritic vascular structure was successfully realized. Figure 9 represents the evolution of vascular engineering, transitioning from inorganic to organic materials, and advancing towards in vitro culture techniques and direct human cell attachment to scaffolds for improved compatibility. Moreover, it showcases the achievement in developing scaffolds that closely mimic dendritic vascular structures through methods such as molding.

As we have seen, the development of bacterial cellulose (BC) for vascular graft applications has yielded substantial advancements. The ability to mimic complex vascular structures and introduce cellular components marks a significant stride toward more successful and compatible grafts. However, it is important to recognize that despite these advancements, we are still faced with several challenges and limitations in fully realizing the potential of BC in vascular grafts. In the following section, we will delve into these challenges, which range from fabrication issues to biocompatibility and regulatory considerations.

## 5. Limitations and Drawbacks of Bacterial Cellulose in Vascular Grafts

Despite the promising prospects of bacterial cellulose (BC) in vascular graft applications, there exist several inherent limitations that need to be addressed. One critical challenge lies in the fabrication process. While the biosynthesis of BC offers environmental sustainability, the process can be time-consuming and achieving uniformity and reproducibility remains a complex task. Physicochemically, BC’s inherent hydrophilicity, though beneficial for some applications, may pose a challenge to cell adhesion. Strategies such as surface modifications have been proposed but ensuring uniform and stable modifications without loss of favorable properties calls for intricate optimization. 

Furthermore, while BC exhibits commendable biocompatibility, the risk of potential inflammatory reactions or immune responses cannot be completely ruled out, and that BC cannot be degraded in vivo is also a problem. For larger grafts or tissues, the lack of an internal vasculature in BC scaffolds can limit the nutrient and oxygen supply, affecting cell survival and functionality. Additionally, regulatory and safety aspects pose considerable challenges, given the concerns about sterilization, batch-to-batch variability, and the long-term stability of BC grafts. Overall, it is essential to investigate further modification and optimization of vascular graft properties.

Herein, there are several strategies that could potentially optimize the production and manufacturing of bacterial cellulose (BC). Engineering solutions such as automation and process control could be utilized to precisely control culture composition and growth conditions, while also facilitating large-scale production. Systems such as these also allow for continuous monitoring and real-time adjustments throughout the production process. The post-processing stages are equally critical. Consistent post-processing procedures, including purification and drying, can contribute to the reproducibility of the final product.

Beyond the production of BC itself, there are opportunities for improvements in the production of BC vascular grafts. Further processing of BC into other materials, such as bio-inks through mechanical or chemical methods, can offer new avenues for the fabrication of vascular grafts. Advanced manufacturing techniques such as bioprinting can then be employed. The utilization of these innovative methods could lead to the production of high-quality, consistent products. This multi-dimensional approach addresses several of the inherent challenges associated with BC production and highlights the importance of engineering principles in optimizing the production of BC for use in vascular graft applications.

## 6. Prospects and Conclusions

The demand for vascular grafts is surging on a global scale, driven largely by an aging population. Small-diameter vessels, in particular, present unique challenges with respect to performance characteristics and material preparation for clinical application. Synthetic vascular grafts offer a promising solution to this growing clinical need, providing a viable alternative to autologous blood vessels. To be deemed effective, these grafts must exhibit several essential properties [114]:(1)Biocompatibility: The grafts must be biocompatible to avoid inciting immune reactions and inflammation. Bacterial cellulose has shown promising results in this respect, by minimizing immune responses and recruitment of immune cells.(2)Mechanical robustness: The grafts should possess mechanical stability and strength commensurate with their intended function.(3)Non-thrombogenic: Grafts must prevent the formation of thrombi and inhibit platelet activation to reduce the risk of blood vessel blockage.(4)Scalable production: A scalable manufacturing process is essential to ensure widespread availability of the grafts while reducing costs.(5)Long-term storage stability: The grafts must maintain their integrity and functionality over an extended storage period, complementing the cost-effectiveness and availability facilitated by mass production.(6)Versatility: Grafts should be flexible enough to meet varying application needs in later stages.

BC is emerging as an ideal material for tissue engineering scaffolds, given its proven biocompatibility, non-thrombogenic nature, mechanical robustness, and suitability for large-scale production and storage. For use in vascular tissue engineering, the scaffolds need to mimic the multilayer structure of the native extracellular matrix, thereby supporting cell attachment, proliferation, and differentiation. While mechanical properties can be fine-tuned through variation in incubation times and drying techniques, further optimization of the porosity and surface properties of bacterial cellulose is needed to enhance cell viability and function.

As previously mentioned, in Section 5., although BC holds great promise, it also comes with certain limitations and challenges. Nonetheless, the intersection of 3D printing technology, biomaterials, biomedicine, and information technology is poised to lead to future breakthroughs in the field. As we continue to make efforts in modifying BC properties to overcome the existing challenges, the prospects of utilizing BC in vascular graft applications remains highly promising and appealing.

## Figures and Tables

**Figure 1 polymers-15-03812-f001:**
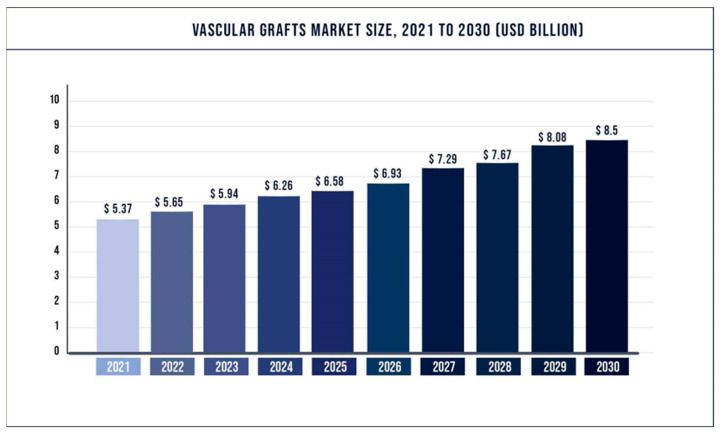
This figure presents a projection of the vascular grafts’ market size from 2021 to 2030, measured in USD Billion. It provides a comprehensive view of the expected growth in the market, highlighting the increasing demand for vascular grafts in the healthcare sector. The data represented in this figure underscores the significant potential for advancements in vascular graft technologies and materials (Data from PRECEDENCERESEARCH).

**Figure 2 polymers-15-03812-f002:**
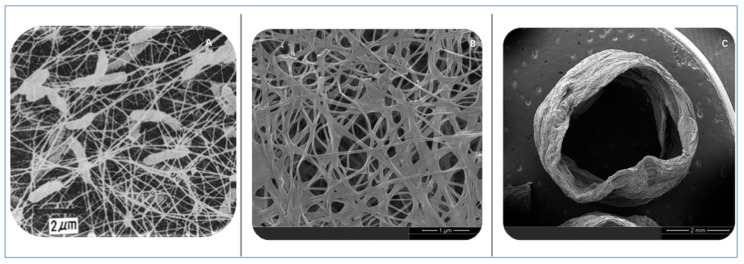
This figure provides a comprehensive overview of the properties and applications of bacterial cellulose. (**A**) Bacterial cellulose being produced by bacteria [17], showcasing the biological origin of this promising material. (**B**) The morphology of lyophilized bacterial cellulose is highlighted, emphasizing its unique three-dimensional structure that contributes to its versatile applications. (**C**) A tubular structure composed of laminated bacterial cellulose is presented, illustrating its potential application in mimicking blood vessel structures. This figure underscores the versatility and potential of bacterial cellulose in various biomedical applications, particularly in vascular graft engineering.

**Figure 3 polymers-15-03812-f003:**
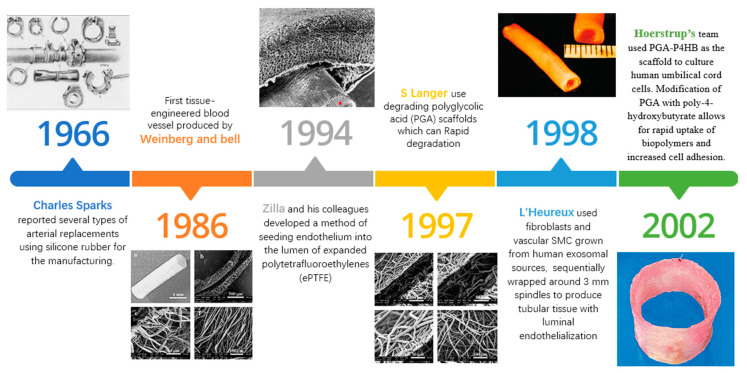
This figure presents a chronological visual journey through the early explorations in artificial blood vessel development. It illustrates the progression and evolution of various strategies over time, providing a foundational understanding of the potential of bacterial cellulose in this field. The figure also highlights the emergence of porous materials as a promising avenue to emulate the extracellular matrix (ECM) and promote dynamic cell growth. The images in the figure above are excerpts from the articles cited [20,22,25,26,28], corresponding to the references in the text above.

**Figure 4 polymers-15-03812-f004:**
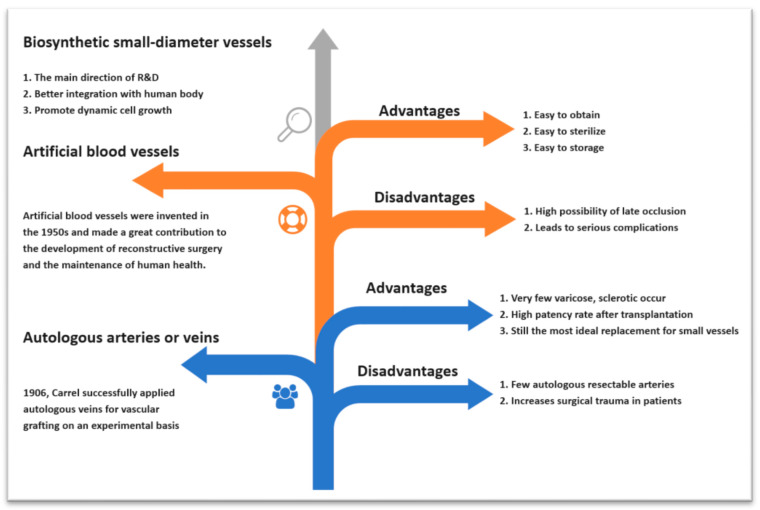
This figure provides a visual representation of the evolution in materials used for vascular grafts. It traces the progression from autologous arteries or veins to artificial blood vessels, and finally to the current focus of research—biosynthetic small-diameter vessels, such as those fabricated from bacterial cellulose. This diagram offers a clear comparison of the advantages and disadvantages associated with each material.

**Figure 5 polymers-15-03812-f005:**
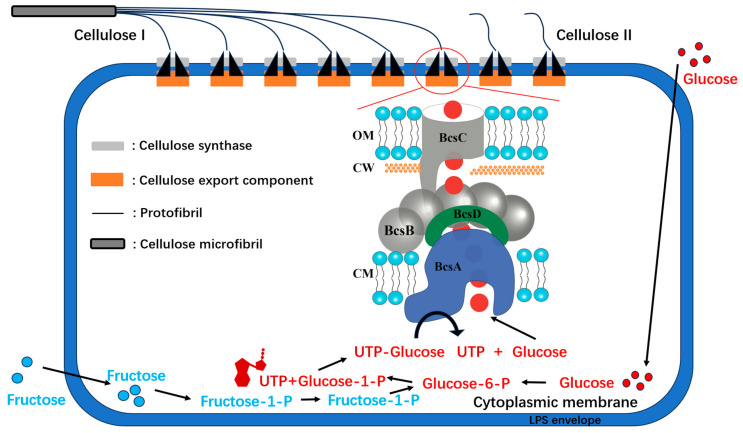
This figure presents a detailed schematic diagram of the cellulose production process in bacteria. The diagram illustrates the transformation of UDP-Glucose into cellulose microfibrils by the enzyme BcsA. These microfibrils are then synthesized into type I cellulose by BcsD and BcsC. The enlarged image provides a detailed view of the cellulose synthase complex in *k. xylinus*, showcasing the intricate processes involved in cellulose synthesis. The diagram is divided into distinct sections, each representing a specific component of the bacterial cell involved in cellulose production: OM represents the outer membrane, CW stands for the cell wall, and CM denotes the cytoplasmic membrane. This illustration provides an in-depth understanding of the bacterial cellulose production process, highlighting the complexity and precision of this natural phenomenon.

**Figure 8 polymers-15-03812-f008:**
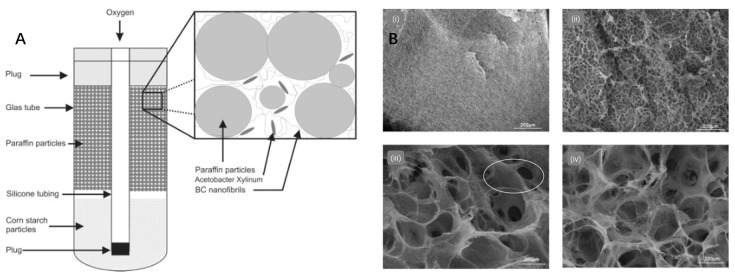
This figure provides a comprehensive view of the impact of different treatment conditions on the pore sizes of bacterial cellulose (BC). (**A**) A schematic representation of the fermentation unit, demonstrating how paraffin granules float in the medium, while starch granules sink. This configuration encourages bacteria to generate a tangled network of BC on and between the pellets, proximal to the silica tube [85]; (**B**) Scanning electron microscope (SEM) imagery detailing the external surfaces of BC tubes produced under varying conditions: (i) BC cultured without any porosity agents, (ii) BC cultured in the presence of potato starch, (iii) BC cultured with paraffin, and (iv) BC cultured with fused paraffin pellets. The circled areas in (iii) highlight regions between the pores [85].

**Figure 9 polymers-15-03812-f009:**
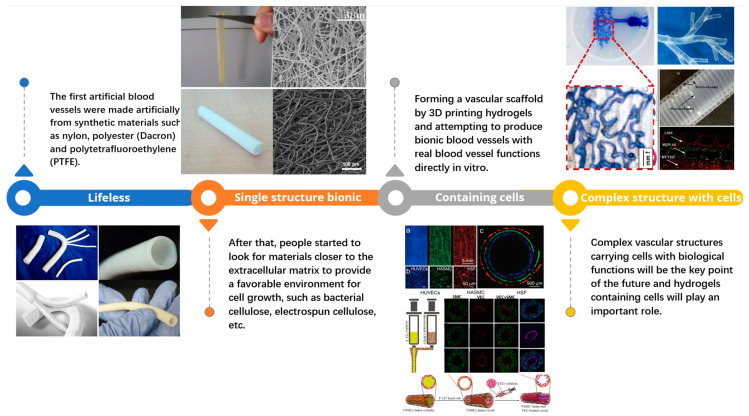
This figure provides a comprehensive overview of the progression in vascular tissue engineering. It is divided into four parts, each representing a significant stage in the evolution of methodologies in this field. Lifeless: Represents the initial phase where inorganic materials were used, demonstrating their limited functionality and compatibility [110]. Single Structure Bionic: Highlights the transition to the use of organic materials that mimic simple vascular structures, showcasing an improved but still limited biological functionality [111]. Containing Cells: Depicts the further development of scaffolds that not only mimic the structure but also support in vitro culture of human cells for enhanced biological functionality [108,112]. Complex Structure with Cells: This section illustrates the current state-of-the-art method where scaffolds can mimic complex dendritic vascular structures and support the growth of different cell types, greatly improving compatibility and functionality [109,113]. The accompanying text and diagrams in each part provide specific examples of advancements at each stage of this evolution, offering a comprehensive understanding of the progress and potential in vascular tissue engineering.

**Table 1 polymers-15-03812-t001:** Different raw materials used for BC production.

Strains	Culture Medium	Culture Time	Capacity	Ref.
*K. xylinus* NUST4.1	Solid corn steep liquor	120 h	6.0 g/L	[54]
*Acetobacter* sp. A9	Standard medium	7 days	Standard medium: 2.2 g/L	[55]
	Production medium		Improved medium 1: 3.8 g/L	
	Optimized medium		Improved medium 2: 15.2 g/L	
*K. xylinus* ATCC 23770	Wheat straw	8 days	8.3 g/L	[56]
*G. xylinus* C3	Industrial wastes	/	8.21 g/L	[57]
*K. xylinus* ATCC 23770	Konjac powder	8–23 days	2.12 g/L	[58]
Gluconacetobacter medellensis	Hestrin–Schramm medium	8 days	4.5 g/L	[59]
*K. xylinus* NBRC 13693	Orange juice,	14 days	Orange juice: 5.9 g/L	[60]
	Pineapple juice,		Pineapple juice: 4.1 g/L	
	Apple juice,		Apple juice: 3.9 g/L	
	Japanese pear,		Japanese pear: 3.5 g/L	
	Grapes		Grape: 1.8 g/L	

**Table 2 polymers-15-03812-t002:** Differences in strength of bacterial cellulose under different culture conditions.

T (°C)	Time (Days)	Culture Medium	Mold	Parameters	Radial (Max)	Axial (Max)	Ref.
28	14	H–S medium	/	ID: 1 mm Wall thickness: 0.7 mm	Load: 0.8 N Stress: / Pressure: 20 Kpa	/	[64]
30	3	Corn steep liquid media	Roux flasks, pellicle of 3 mm	ID: 5 mm OD: 10 mm	Load: / Stress: 0.16 Mpa Pressure: /	/	[65]
28	7	H–S medium	silicone tubes	ID: 6 mm Wall thickness: 1 mm	Load: / Stress: 0.59 Mpa Pressure: /	Load: / Stress: 0.37 Mpa	[66]
30	4–7	Green tea medium with Kombucha	Double-silicone tubes	ID: 3 mm Wall thickness: 3 mm	Load: 1.22–1.6 N Stress: / Pressure: 29.2 ± 3.4 Kpa	Load: 1.3–2.8 N Stress: /	[68]
28	3–18	H-S medium	PDMS	/	Load: 0.38–1.47 N Stress: 0.04–0.19 Mpa Pressure: /	Load: 0.63–2.09 N Stress: 0.48–0.81 Mpa	[67]

## Data Availability

Not applicable.

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
