# Peer review of "Bacterial Nanocellulose Hydrogel: A Promising Alternative Material for the Fabrication of Engineered Vascular Grafts"

_polymers, 2023, doi:10.3390/polym15183812_

Round 1
Reviewer 1 Report
The manuscript is very well written and organizer review. I have only some suggestions before acceptance:
It is worth comparing the discussed properties, especially the mechanical ones, with those characteristics for commercial, non-BC-made, artificial, and natural blood vessels.
1. Line 30: should be (Figure 1)
2. Line 31: CAGR - explain all abbreviations when they are used for the first time (check all in the manuscript)
3. Figure 2: enlarge A, B, and C marks
4. In Table 1: explain what constitutes the standard medium, improved medium 1, and improved medium 2 (ref. [55])
5. Figure 6A: remove numbers (or explain them)
6. Line 321: indicate what kind of mechanical properties were changed
7. Line 379: Bacillus cereus: use italic
8. check ref 115 - is it cited in the text?
9. Some references should be improved according to the journal guide ([4], [9], etc., Publisher: Publisher Location, Country, Year /or/ journal name, issue, volume)
Author Response
Thank you for your valuable suggestion to compare the discussed properties. Your suggestion is indeed insightful. However, there is a lack of clear data for the commercial, non-BC-made, artificial and natural smaller blood vessels. To the best of our knowledge, there are no successfully commercialized small-diameter blood vessels made of other materials. In addition, nature blood vessels often use mmHg to indicate their burst pressure, but there is scant test data on burst pressure for BC vessels. Hence, we couldn’t provide a comparative analysis in this regard due to these disparities. We appreciate your understanding and your insightful comments, which have indeed strengthened our future research.
- Line 30: should be (Figure 1)
Thank you very much for your suggestion. We have made corrections at the original location and marked them in red.
- Line 31: CAGR - explain all abbreviations when they are used for the first time (check all in the manuscript)
Thank you very much for your suggestion. We have made corrections at the original location and marked them in red.
- Figure 2: enlarge A, B, and C marks
Thank you very much for your suggestion. We have made corrections at the original location and marked them in red.
- In Table 1: explain what constitutes the standard medium, improved medium 1, and improved medium 2 (ref. [55])
Thank you very much for your suggestion. Indeed, this was a point that was not clearly expressed. We have now expanded on this in the text, discussing the differences among the three culture media in the revised manuscript.
- Figure 6A: remove numbers (or explain them)
This is a very good suggestion, which would indeed help readers understand the structure of the device. We have now added the corresponding explanations to these numbers in the text.
- Line 321: indicate what kind of mechanical properties were changed
Thank you very much for your suggestion. We have made corrections at the original location and marked them in red.
- Line 379: Bacillus cereus: use italic
Thank you very much for your suggestion. We have made corrections at the original location and marked them in red.
- check ref 115 - is it cited in the text?
Thank you very much for your suggestion. We have made corrections at the original location and marked them in red.
- Some references should be improved according to the journal guide ([4], [9], etc., Publisher: Publisher Location, Country, Year /or/ journal name, issue, volume)
Thank you very much for your suggestion. We have made corrections at the original location.
Reviewer 2 Report
The manuscript systematically discussed recent progress in fabricating engineered vascular grafts using bacterial nanocellulose, and demonstrated the efficacy of bacterial cellulose hydrogel as a biomaterial for synthetic vascular grafts, specifically for stimulating angiogenesis and neovascularization. The discuss is reasonable and acceptable, from simple to profound, from point to plane. Thus, it can be accepted in present form.
Author Response
Thank you for your recognition of our work!
Reviewer 3 Report
The authors summarized the recent progress of BC in creating blood vessel scaffold. I don't see significant flaw for the review content. I would like to invite the authors to elaborate more on section 5 limitations part: In terms of the time consuming fabrication and reproducibility, what would be the potential technical solutions?
Author Response
Thanks to your suggestion, we have added the following description to the text and it reads:
Herein, there are several strategies that could potentially optimize the production and manufacturing of bacterial cellulose (BC). Engineering solutions such as automation and process control could be utilized to precisely control culture composition and growth conditions, while also facilitating large-scale production. Systems like these also allow for continuous monitoring and real-time adjustments throughout the production process. The post-processing stages are equally critical. Consistent post-processing procedures, including purification and drying, can contribute to the reproducibility of the final product.
Beyond the production of BC itself, there are opportunities for improvements in the production of BC vascular grafts. Further processing of BC into other materials, such as bio-inks through mechanical or chemical methods, can offer new avenues for the fabrication of vascular grafts. Advanced manufacturing techniques like bioprinting can then be employed. The utilization of these innovative methods could lead to the production of high-quality, consistent products. This multi-dimensional approach addresses several of the inherent challenges associated with BC production and highlights the importance of engineering principles in optimizing the production of BC for use in vascular graft applications.